# Explainable Fake News Detection With Large Language Model via Defense Among Competing Wisdom

## ABSTRACT

Most fake news detection methods learn latent feature representations based on neural networks, which makes them black boxes to classify a piece of news without giving any justification. Existing explainable systems generate veracity justifications from investigative journalism, which are debunking delayed and low efficiency. Recent studies simply assume that the justification is equivalent to the majority opinions expressed in the wisdom of crowds. However, the opinions typically contain some inaccurate or biased information since the wisdom of crowds is uncensored. To detect fake news from a sea of diverse, crowded and even competing narratives, in this paper, we propose a novel defense-based explainable fake news detection framework. Specifically, we first propose an evidence extraction module to split the wisdom of crowds into two competing parties and respectively detect salient evidences. To gain concise insights from evidences, we then design a prompt-based module that utilizes a large language model to generate justifications by inferring reasons towards two possible veracities. Finally, we propose a defense-based inference module to determine veracity via modeling the defense among these justifications. Extensive experiments conducted on two real-world benchmarks demonstrate that our proposed method outperforms state-of-the-art baselines in terms of fake news detection and provides high-quality justifications.

## KEYWORDS

Fake News Detection, Explainable, Large Language Model, Competition in Wisdom, Defense-based Inference

**ACM Reference Format:**

Anonymous Author(s). 2023. Explainable Fake News Detection With Large Language Model via Defense Among Competing Wisdom. In *Proceedings of the Web Conference 2024 (WWW '24), May 13–17, 2024, Singapore*. ACM, New York, NY, USA, 11 pages. https://doi.org/XXXXXXX.XXXXXXX

## 1 INTRODUCTION

The proliferation of fake news on social media has become a remarkable concern, leading to detrimental effects on individuals and society. For example, during the global COVID-19 pandemic, a piece of spurious news claiming that "*the COVID-19 vaccine can induce serious side effects and potentially result in death*[1]" attracted the public's attention, leading to people's hesitancy and resistance towards vaccine uptake, and thus seriously impacted the virus containment effort and overwhelmed healthcare systems around the

[1]https://www.bbc.com/news/53525002

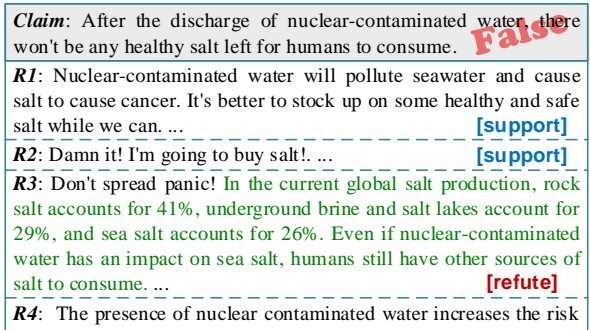

**Figure 1: A false claim from the Sina Weibo. The comparison of informativeness and soundness between two competing parties serves as an indicator of veracity.**

world. Fortunately, as the truth was consistently justified by the official media and investigative journalism, the public recognized the claim as fake. This indicates the positive role of solid justification in restricting the social harmfulness caused by fake news. However, relying solely on investigative journalism to enable the public to detect fake news is not a realistic approach. Such a labor-intensive and time-consuming manner limits the coverage and is debunking delayed. Thus, it is urgent to develop automated methods to detect fake news and provide clear justifications timely.

Most previous works for detecting fake news focus on incorporating various information to learn the latent features of fake news via deep networks, e.g., credibility [25], stances [20], propagation patterns [28], extra knowledge [10], etc. Despite their success in detecting, limited by their black-box nature, they are unable to provide any justification, which is essential to the public. To address this problem, some studies are dedicated to explainable fake news detection (EFND) that aims to generate veracity prediction and explanations at the same time [14]. Many works provide their interpretability by highlighting salient words, phrases, or sentences in relevant reports via attention mechanism [21, 26, 29, 35]. However, these works only uncover regions with high contributions for the final prediction, lacking intuitive and credible explanations. As human justification brings great improvement in the veracity prediction [1], some works take debunked reports into consideration to generate explanations. Atanasova et al. [3] obtains explanations by summarizing from debunked reports, but suffers from debunking delayed and low efficiency. To mitigate this problem, motivated by the effectiveness of the wisdom of crowds in fact-checking [2], Yang et al. [38] assumes the majority of opinions expressed in the raw reports are equivalent to the justifications and extracts explanations from them. However, unverified raw reports typically contain inaccurate or biased information since the wisdom of crowds is

uncensored. The ill-considered assumption leads to misleading results that are biased towards the majority opinion in raw reports. Therefore, it remains a challenge to effectively leverage the rich wisdom expressed in raw reports to support EFND.

Recent detection in the field of stance detection implicitly suggests that the different insights in various raw reports are crucial signals in the quest for truth [20, 37]. Inspired by it, we propose to split the wisdom into two distinct parties, which allows the detection to rely on the quality of wisdom rather than its quantity. Take a concrete example, as shown in Figure 1, there are two competing parties to the claim. For the supporting party, R1 and R4 both briefly discuss the risk of salt leading to cancer, and R2 echoes the claim without additional information. In contrast, R3 provides the refuting party with detailed evidence to illustrate its unique viewpoint, which is solid and persuasive. Based on the observation, we assume that the reports indicating truthfulness could exhibit higher quality of informativeness and soundness compared to those conveying inaccurate information. As a result, the veracity of news can be ascertained through a comparative analysis between two competing parties. Therefore, how to effectively split the wisdom into two parties from raw reports and then capture their quality divergence, is a critical problem for enhancing explainable fake news detection.

To deal with the above issues, we propose a defense-based explainable fake news detection framework, which strives to capture the divergence between the competing wisdom reflected in raw reports and pursue the veracity of claims in a defense-like way. Specifically, we first propose an evidence extraction module to split the wisdom of crowds into two competing parties, from which we detect salient evidences, respectively. Since the wisdom of each competing party is massive, it is formidable to identify the divergence by directly using the competing evidences, thereby raising the demand for streamlined summarization. Inspired by the dominating performance of large language models (LLMs) [22, 23, 31, 32], based on the respective evidences, we then design a prompt-based module to generate justifications by inferring reasons towards two possible veracities. Benefiting from the powerful reasoning and generating capacity of LLMs, we obtain the summarized wisdom of both parties in natural language, allowing for explicit comparisons. Finally, to capture the winner in quality comparisons, namely the party indicating the veracity of the claim, we propose a defense-based inference module to determine veracity by modeling the defense among these justifications. In this manner, the final justification for the verdict is adaptively selected from these justifications.

We evaluate our proposed framework based on two real-world fake news detection benchmarks. The results not only show that our framework outperforms strong fake detection baselines by a large margin, but also provide high-quality explanations. Our main contributions are summarized as follows:

- We develop a novel defense-based framework to effectively utilize the rich competing wisdom naturally contained in raw reports, mitigating the majority bias problem from which existing works suffer.
- By integrating the powerful reasoning capabilities of LLMs, our model can derive explanations comparable to those of human experts without any debunked reports' supervision.

- We achieve new state-of-the-art results on two real-world explainable fake news detection datasets and demonstrate the quality of the explanations with extensive analyses.

## 2 RELATED WORK

### 2.1 Explainable Fake News Detection

Much effort has been devoted to investigating the field of explainable fake news detection in previous studies. To bring about some explainability, some works explore attention mechanisms to highlight salient phrases [26, 35], news attributes [36], and suspicious users [18]. In order to gain more human-readable explanations, some works capture the salient sentences as explanations via sentence-level attention [19, 21, 29]. However, these methods merely uncover regions with high contributions to the final veracity prediction rather than view the explanation generation as a dependent task. To address it, Kotonya and Toni [15] regards the explanation generation task as a pre-trained extractive-abstractive summarization task, independent of the veracity prediction. Atanasova et al. [3] formats the EFND task as a multi-task learning problem, and then trains a joint model to solve the veracity prediction and summarize the explanation based on external debunked reports collected from fact-checking websites. However, the debunking of claims is a labor-intensive and time-consuming process. Such a heavy reliance on debunked reports results in coverage limitations and debunked delays, which restricts its practical application. To alleviate this problem, Yang et al. [38] merely employs the debunked report as a supervised signal in training, and concentrates on the majority opinions from crowds expressed in relevant raw reports to aid in prediction and extract evidence. However, it ignores the inaccurate and biased information in unverified reports, causing a misleading result. Therefore, by splitting the wisdom of crowds into two competing parties and introducing a defense-like strategy, we capture and leverage the divergence between the competing wisdom to reach a verdict on the claim.

### 2.2 Large Language Model in Fake News Detection

Recently, large language models have been proven excellent ability in multiple classification and reasoning tasks [5, 22–24, 32]. Unfortunately, the huge training cost prevents LLMs from keeping up with the latest information, which restricts the application of LLMs in the field of fake news detection with high real-time requirements. Several studies experimentally demonstrate there is a gap between LLMs and fine-tuned small language models (SLMs) like BERT [9], but also indicate that LLMs hold great potential for detecting fake news [11, 16, 30]. Li et al. [16] proposes a step-by-step framework consisting of a set of plug-and-play modules to facilitate fact news detection. It achieves promising results in a zero-shot setting by purely prompting LLMs with external retrieved knowledge. Hu et al. [11] find that LLMs are suboptimal at veracity judgment but good at analyzing contents. It thus trains the small language model to adaptively acquire insights from LLM-generated rationales in a distillation framework. Cheung and Lam [7] uses LoRA tuning [12] to train a LLaMA-based [31] detector with external retrieved knowledge. However, these methods merely concentrate on the detection of fake news and lack the ability to generate an explanation.

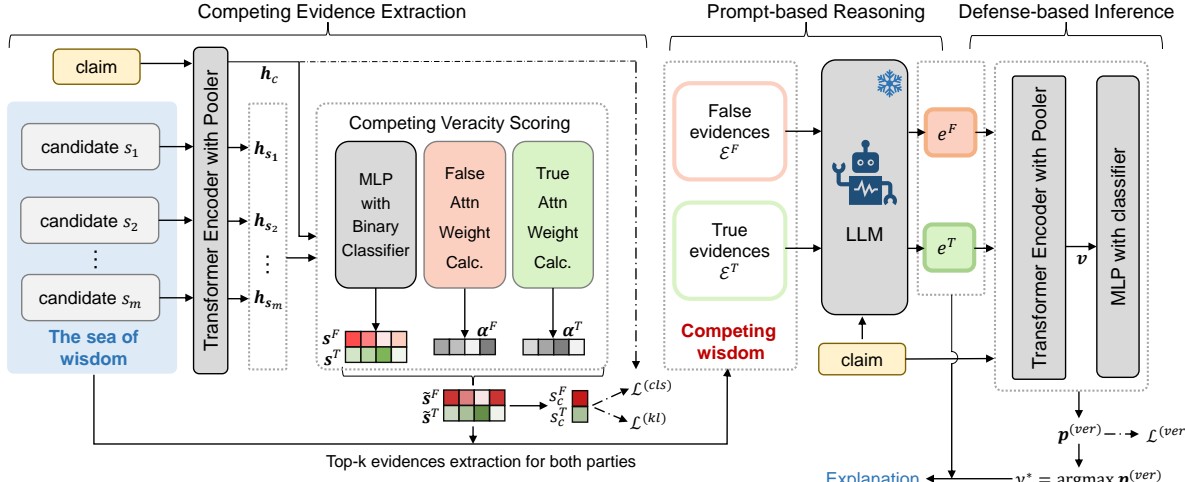

**Figure 2: An overview of the proposed LLM-equipped defense-based explainable fake news detection (L-Defense) framework.**

Also, the methods incorporating external retrieved knowledge still suffer from the majority bias problem. In contrast, we employ LLMs to generate justifications towards two possible veracities based on respective evidences. The derived competing justifications are used to detect veracity via a defense-like process.

## 3 PROPOSED APPROACH

This section begins with a task definition of explainable fake news detection (EFND). Then, we present our **L**LM-equipped **D**efense-based **E**xplainable **F**ake **N**ews **D**etection (L-Defense) framework (as in Figure 2) consisting of three components: a competing evidence extractor (§3.1), a prompt-based reasoning module (§3.2), and a defense-based inference module (§3.3).

*Task Definition.* Given a news claim $c$ associated with a veracity label $y$, and its relevant raw reports $\mathcal{D} = \{d_i\}_{i=1}^{|\mathcal{D}|}$, where each $d_i = (s_{i,1}, s_{i,2}, \ldots, s_{i,|d_i|})$ denotes a relevant report consisted of a sequence of sentences, EFND aims to predict a veracity label $y^*$ from {false, true, . . . } of claim $c$ and provide an explanation $e$ regarding the prediction.

## 3.1 Competing Evidence Extraction

In this subsection, we aim to split the sea of wisdom into two competing parties, and extract respective evidences for subsequent comparison. As the only available supervised data is the veracity label of the claim, we conduct a temporary veracity prediction to detect salient evidences for both parties via a veracity scoring module. Each claim is assigned a temporary veracity label in {false, half, true}, where "half" indicates that the claim contains both true and false aspects.

*3.1.1 Claim and Evidence Representation.* Since a report may contain evidences indicating different veracities, we disaggregate the reports into individual sentences, resulting in a corpus of candidate evidence sentences $\mathcal{S} = \{s_j\}_{j=1}^{m}$, where $m = \sum_{d_i \in \mathcal{D}} |d_i|$. We thereby adopt a vanilla pre-trained transformer encoder [33] to generate representations of claim and candidate evidences. Formally, a Transformer-Encoder is fed with a claim or a candidate,

$$\boldsymbol{h}_c = \text{Pool}(\text{Transformer-Enc}(c; \theta^{(ec)})), \quad (1)$$

$$\boldsymbol{h}_{s_j} = \text{Pool}(\text{Transformer-Enc}(s_j; \theta^{(ec)})), \quad (2)$$

where $\text{Pool}(\cdot)$, defined in [9], collects the resulting of [CLS] to denote a contextualized representation for the sequence, $\theta^{(ec)}$ denotes the learnable parameters of transformer encoder.

*3.1.2 Competing Veracity Scoring.* In order to detect salient evidences for both parties, we propose to assign two veracity scores for each candidate, namely the "false score" and the "true score", which represent the level of support for the claim being true or false, respectively. Naturally, the two scores can be used for ranking to extract the top-$k$ "false evidences" and "true evidences" in inference.

To gain the veracity scores, we first calculate a pair of complementary scores for *true* and *false* for each candidate. Borrowing common practices from the natural language inference (NLI) area, we apply an interactive concatenation [4, 27] to the pair of claim and candidate, and then perform a neural binary classifier. Formally, we adopt the interactive concatenation written as

$$\boldsymbol{u}_j = [\boldsymbol{h}_c; \boldsymbol{h}_c \times \boldsymbol{h}_{s_j}; \boldsymbol{h}_c - \boldsymbol{h}_{s_j}; \boldsymbol{h}_{s_j}], \quad (3)$$

where $\boldsymbol{u}_j$ is used to represent the semantic relationship between the claim and candidate. Then, a two-way classifier is applied to $\boldsymbol{u}_j$ and produces a two-dimensional categorical distribution corresponding to the *false* and *true* veracity probabilities respectively, i.e.,

$$\boldsymbol{p}_j = P(z^{(vp)}|\boldsymbol{u}_j; \theta^{(vp)}) \triangleq \text{softmax}(\text{MLP}(\boldsymbol{u}_j; \theta^{(vp)})) \in \mathbb{R}^2, \quad (4)$$

where $\text{MLP}(\cdot)$ stands for a multi-layer perceptron, and $\theta^{(vp)}$ is its learnable parameters. In this way, the *false* and *true* probabilities,

$$s_j^F = \boldsymbol{p}_j^1, \; s_j^T = \boldsymbol{p}_j^2, \quad (5)$$

can serve as veracity scores of the candidate to perform false and true candidate ranking, respectively.

Nevertheless, merely employing the complementary scores of each candidate for the ranking purpose is suboptimal, as not every

sentence carries valuable information or contributes significantly to the final veracity prediction [3, 38]. To address this limitation, we calculate each candidate's attention weights for the claim under two competing veracities, enabling a more precise ranking of the evidences. Formally, we present a concatenation-based attention weight calculation module to gain the false (true) attention weight,

$$\alpha_j^v = \frac{\exp(\text{MLP}([\boldsymbol{h}_c; \boldsymbol{h}_{s_j}]; \theta^{(attn-v)}))}{\sum_k \exp(\text{MLP}([\boldsymbol{h}_c; \boldsymbol{h}_{s_k}]; \theta^{(attn-v)}))}, \tag{6}$$

where $v$ alternates between F (false) and T (true), $\alpha_j^v$ denotes false score $\alpha_j^F$ when $v$ is F, and $\alpha_j^v$ denotes true score $\alpha_j^T$ when $v$ is T. As a result, the final false (true) score of candidate evidence used for ranking is

$$\tilde{s}_j^v = \alpha_j^v s_j^v. \tag{7}$$

Based on the competing veracity scores of candidate evidences, we can naturally obtain the competing veracity scores of a claim by

$$s_c^v = \sum \alpha_j^v \tilde{s}_j^v, \tag{8}$$

which can be used to judge the veracity of a claim.

*3.1.3 Training and Inference.* Since the only available supervised data is the veracity label of the claim, we employ two loss objectives to conduct the temporary veracity prediction, taking into consideration the extraction of competing evidences.

*Training.* The primary objective of this module is to rank and extract two competing sets of evidence using the competing veracity scores. To achieve this, we introduce a soft objective that considers the competing scores, utilizing the Kullback–Leibler (KL) divergence, i.e.,

$$\mathcal{L}^{(kl)} = \text{KL}(\boldsymbol{p} || \boldsymbol{p}_c), \tag{9}$$

$$\text{where } \boldsymbol{p} = \begin{cases} (1.0, 0.0) & y^t = \text{false} \\ (0.5, 0.5) & y^t = \text{half} \\ (0.0, 1.0) & y^t = \text{true} \end{cases} \tag{10}$$

where $\boldsymbol{p}_c = (s_c^F, s_c^T) \in \mathbb{R}^2$ is derived from Eq.(8), $y^t$ denotes the temporary veracity label. However, this soft KL loss is not suitable for the inference purpose and its connection with the claim is not strong enough. Hence, we further define a neural classifier for the temporary three-categorical veracity prediction as

$$\begin{aligned} \boldsymbol{p}^{(cls)} &= P(z^{cls} | \boldsymbol{h}_c; \boldsymbol{p}_c; \theta^{(cls)}) \\ &\triangleq \text{softmax}(\text{MLP}([\boldsymbol{h}_c; \boldsymbol{p}_c]; \theta^{(cls)})) \in \mathbb{R}^3. \end{aligned} \tag{11}$$

Next, the classification objective to train this module is

$$\mathcal{L}^{(cls)} = - \sum_{\mathcal{D}} \log \boldsymbol{p}^{(cls)}_{[\hat{y}^t = y^t]}, \tag{12}$$

where $\boldsymbol{p}^{(cls)}_{[\hat{y}^t = y^t]}$ denotes fetching the probability value corresponding to the temporary veracity label $y^t$.

We train the learnable parameters in our proposed extraction module towards a linear combination of the two losses, i.e.,

$$\mathcal{L}^{(ext)} = \gamma \mathcal{L}^{(cls)} + (1 - \gamma)\mathcal{L}^{(kl)}, \tag{13}$$

where $\gamma$ is the trade-off weight.

*Inference.* After optimizing the extraction module w.r.t $\mathcal{L}^{(ext)}$, $s_j^F$ and $s_j^T$ from Eq.(5) produced in inference can be used as ranking basis. For all candidate evidences, they will be ranked twice based on $s_j^F$ and $s_j^T$ respectively. The sets of top-$k$ false and true evidences, i.e., $\mathcal{E}^F$ and $\mathcal{E}^T$, are then prepared for the prompt-based reasoning module described in the subsequent section.

## 3.2 Prompt-based Reasoning with LLM

To effectively leverage the divergence contained in massive competing evidences on informativeness and soundness, we develop a prompt-based module for a further filter and summarization. Motivated by the remarkable abilities of LLMs in reasoning [22, 32], we engage an LLM to conduct abductive reasoning to explain why the claim is false or true based on the extracted evidence sets and a given prior veracity label.

Given a claim $c$, a prior label $\tilde{y}^v$, and an evidences set $\mathcal{E}^v$, to prompt the large language model in uniform language modality, we curate a template $T$ that consists of a triplet $\{c, \tilde{y}^v, \mathcal{E}^v\}$. We prompt the LLM with it to generate an explanation $e^v$ that elicits the reasoning knowledge about how to infer the veracity label $\tilde{y}^v$ based on the interplay of the claim $c$ and the veracity-oriented evidence $\mathcal{E}^v$. Specifically, we design $T$ as:

" *Given a claim: [c], a veracity label [$\tilde{y}^v$], please give me a streamlined rationale associated with the claim, for how it is reasoned as [$\tilde{y}^v$]. Below are some sentences that may be helpful for the reasoning, but they are mixed with noise: [$\mathcal{E}^v$].* "

The reasoning is performed for both *false* and *true* and thus two reasoning texts are obtained. As detailed previously, the evidence set which is consistent with the actual veracity of the claim brings more information and is more reasonable than the competing one. Thus, the LLM prefers to generate solid reasoning in favor of it, while providing weak reasoning with inaccurate information for its competitor. In this manner, the two LLM-generated veracity-oriented reasoning $e^F$ and $e^T$, which can be viewed as two explanations to clarify its relevant veracity label, will always possess a relative strength in confidence, greatly facilitating the detection of fake news.

As detailed in related work, LLMs can provide desirable multiperspective rationales but still underperform the basic fine-tuned small language models [11]. Therefore, based on the generated explanations, we further propose a defense-based inference module with an SLM.

## 3.3 Defense-based Inference

With the news claim $c$ and two veracity-oriented explanations derived from LLM, we develop a defense-based fake news detector. This detector aims to discern the relative strength of the two explanations from their defense, ultimately providing the veracity verdict. Concretely, we concatenate the three texts and feed them into a pre-trained Transformer encoder for a contextual representation. Formally,

$$\boldsymbol{x} = \text{Transformer-Enc}([c; [SEP]; e^F; [SEP]; e^T]; \theta^{(ed)}), \tag{14}$$

where [SEP] denotes the special separate token defined in [9]. Due to the stacked transformer encoders, this representation can effectively capture the semantic differences and connections between

these three texts. Then, we define a classifier on the top of their rich representation for veracity prediction,

$$p^{(ver)} = P(z^{(ver)}|x; \theta^{(ver)})$$
$$\triangleq \text{softmax}(\text{MLP}(x; \theta^{(ver)})) \in \mathbb{R}^N, \quad (15)$$

where $N$ denotes the number of labels.

The training objective of the veracity prediction task is written as:

$$\mathcal{L}^{(ver)} = -\sum_{\mathcal{D}} \log p^{(ver)}_{[\hat{y}=y]}, \quad (16)$$

where $p^{(ver)}_{[\hat{y}=y]}$ denotes fetching the probability value corresponding to the veracity label $y$.

The inference procedure can be simply written as

$$y^* = \arg\max p^{(ver)}. \quad (17)$$

Based on the model's predictions, we select the corresponding explanation as the final explanation $e$, which can be written as:

$$e = \begin{cases} e^F & y^* = \text{false} \\ e^T & y^* = \text{true} \\ e^F; e^T & y^* = \text{half} \end{cases} \quad (18)$$

Especially, in the case of a "half" prediction, to help users understand the false aspect and the true aspect of the claim, we utilize a template to concatenate the two explanations and present both explanations simultaneously. This approach ensures that the final explanation aligns with the predicted veracity label.

## 4 EXPERIMENT

In this section, we evaluate L-Defense on two real-world explainable fake news detection benchmarks, and verify the model's effectiveness (§4.1) and explainability (§4.2). Then, we conduct an extensive ablation study in §4.3 to verify the significance of each proposed module. Lastly, in §4.4, we make comprehensive analyses to show how our proposed model brings improvement.

*Datasets.* We assessed the proposed approach on two explainable datasets, i.e., *RAWFC* and *LIAR-RAW* [38], whose statistics are listed in Table 1. *RAWFC* contains the claims collected from Snopes[2] and relevant raw reports by retrieving claim keywords. For *LIAR-RAW*, it is extended from the public dataset LIAR-PLUS [1] with relevant raw reports, containing fine-grained claims from Politifact[3]. Note that we do not use any debunked justifications in the datasets for both training and inference.

*Training Setups.* We initialize the transformer encoder in the first extraction module (§3.1) with RoBERTa$_{base}$ [17] for the temporary veracity prediction, and set $k = 10$ to extract evidences. As for the LLM used in §3.2, we alternate between ChatGPT [23] and LLaMA2$_{7b}$ [32]. The former refers to a widely used LLM developed by OpenAI, specifically utilizing the "gpt-3.5-turbo" version, and the latter is a smaller yet powerful LLM created by Meta AI. To obtain the final prediction, we initialize the transformer encoder in §3.3 with the RoBERTa$_{large}$. Please refer to Appendix A for more training details.

[2]https://www.snopes.com/
[3]https://www.politifact.com/

Table 1: Summary statistics of datasets. The numbers range from 0 to 5, representing the increasing veracity labels {pants-fire, false, barely-true, half-true, mostly-true, true}. "ALL" means the total number, and $|\mathcal{S}|_{avg}$ denotes the average number of sentences associated with each claim.

| | | 0 | 1 | 2 | 3 | 4 | 5 | ALL | $|\mathcal{S}|_{avg}$ |
|---|---|---|---|---|---|---|---|---|---|
| LIAR-RAW | train | 812 | 1,985 | 1,611 | 2,087 | 1,950 | 1,647 | 10,065 | 62 |
| | eval | 115 | 259 | 236 | 244 | 251 | 169 | 1,274 | 80 |
| | test | 86 | 249 | 210 | 263 | 238 | 205 | 1,251 | 96 |
| RAWFC | train | - | 502 | - | 532 | - | 555 | 1,589 | 154 |
| | eval | - | 66 | - | 67 | - | 67 | 200 | 156 |
| | test | - | 66 | - | 67 | - | 67 | 200 | 157 |

Table 2: Veracity prediction results on RAWFC and LIAR-RAW. †Resulting numbers are reported by Yang et al. [38], and the results of FactLLaMA are taken from the original paper. The bold numbers denote the best results in each fine-grained genre while the underlined ones are state-of-the performance.

| | RAWFC | | | LIAR-RAW | | |
|---|---|---|---|---|---|---|
| | P | R | macF1 | P | R | macF1 |
| *Traditional approach* | | | | | | |
| dEFEND [29]† | 44.93 | 43.26 | 44.07 | 23.09 | 18.56 | 17.51 |
| SBERT-FC [15]† | 51.06 | 45.92 | 45.51 | 24.09 | 22.07 | 22.19 |
| GenFE [3]† | 44.29 | 44.74 | 44.43 | 28.01 | 26.16 | 26.49 |
| GenFE-MT [3]† | 45.64 | 45.27 | 45.08 | 18.55 | 19.90 | 15.15 |
| CofCED [38]† | **52.99** | **50.99** | **51.07** | **29.48** | **29.55** | **28.93** |
| *LLM-based approach* | | | | | | |
| LLaMA2$_{claim}$ | 37.30 | 38.03 | 36.77 | 17.11 | 17.37 | 15.14 |
| ChatGPT$_{full}$ | 39.48 | 45.07 | 39.31 | **29.64** | 23.57 | 21.90 |
| ChatGPT$_{claim}$ | 47.72 | 48.62 | 44.43 | 25.41 | **27.33** | **25.11** |
| FactLLaMA [7] | 53.76 | 54.00 | 53.76 | 32.32 | 31.57 | 29.98 |
| FactLLaMA$_{know}$ [7] | **56.11** | **55.50** | **55.65** | **32.46** | **32.05** | **30.44** |
| *Ours* | | | | | | |
| L-Denfense$_{LLaMA2}$ | 60.95 | 60.00 | 60.12 | **31.63** | 31.71 | **31.40** |
| L-Denfense$_{ChatGPT}$ | **61.72** | **61.01** | **61.20** | 30.55 | **32.20** | 30.53 |

### 4.1 Evaluations on Veracity Prediction

*Baselines.* We compare our L-Defense with two categories, traditional non-LLM-based approaches and LLM-based approaches. *Traditional category* contains: 1) **dEFEND** [29]; 2) **SBERT-FC** [15]; 3) **GenFE** and **GenFE-MT** [3]; 4) **CofCED** [38]. And *LLM-based category* contains: 5) **LLaMA2$_{claim}$** (7b version) [32] prompts with the news claim to directly generate a veracity prediction and corresponding explanation; 6) **ChatGPT$_{claim}$** [23], which is similar to LLaMA2$_{claim}$; 7) **ChatGPT$_{full}$** [23] prompts with the claim and all related reports, and the absence of LLaMA2$_{full}$ is that the 7b model struggles to produce consistent output after processing such lengthy inputs; 8) **FactLLaMA** [7] leverages the LORA tuning [12] to supervised fine-tunes a LLaMA$_{7b}$ with the claims; 9) **FactLLaMA$_{know}$**, compared with FactLLaMA, fed with external relevant evidence retrieved from search engines.

The veracity prediction results of competitive approaches and ours on the two benchmarks are shown in Table 2. Following prior

works [38], we adopt macro-averaged precision (P), recall (R), and F1 score (macF1) to evaluate the performance. It is observed that our proposed L-Defense is able to achieve state-of-the-art or competitive performance on the two datasets.

For the traditional approaches, most of them underperform $\text{ChatGPT}_{\text{claim}}$, demonstrating the potential of LLMs in fake news detection. For the first three LLM-based approaches without any tuning, $\text{ChatGPT}_{\text{claim}}$ achieves the best results. $\text{LLaMA2}_{\text{claim}}$ loses as its model size is significantly smaller than that of ChatGPT. And a possible reason why $\text{ChatGPT}_{\text{full}}$ loses is that the LLM is easily biased by the massive input reports. Despite the good performance of $\text{ChatGPT}_{\text{claim}}$, it falls short when compared to the best method in the traditional approach, namely CofCED. By contrast, the fine-tuned LLM-based model, i.e., FactLLaMA and $\text{FactLLaMA}_{\text{know}}$ achieves the best results in addition to our proposed model. This indicates that simply utilizing LLMs for inference yields limited results, while carefully considering how to further leverage LLMs can lead to improved performance.

Our proposed model makes use of LLM as a reasoner in a novel defense-based framework, and then achieves excellent results on veracity prediction. The improvement is especially significant on RAWFC. Compared with the $\text{FactLLaMA}_{\text{know}}$, both versions of our model achieved at least a 4% enhancement across all metrics. And on LIAR-RAW, although our model achieves inferior results on precision, both variants consistently outperform in terms of macF1. Furthermore, in comparison with ours LLaMA2 variant, ours ChatGPT variant achieves slightly superior results on RAWFC while maintaining competitiveness on LIAR-RAW. This suggests that our model does not prioritize the size of the LLM component.

## 4.2 Evaluations on Explanation

*Evaluation Metrics.* For the evaluation of explanations, traditional automated evaluation metrics are inadequate to assess the output results of LLMs [5]. Fortunately, Chen et al. [6] demonstrates that ChatGPT excels in assessing text quality from multiple angles, even in the absence of reference texts. Also, some works reveal that ChatGPT evaluation produces results similar to expert human evaluation [8, 13]. Therefore, we engage ChatGPT to evaluate the quality of explanations based on four metrics which have been widely employed in human evaluation [34, 38]: *misleadingness*, *informativeness*, *soundness*, and *readability*. A 5-point Likert scale was employed, where 1 represented the poorest and 5 the best in addition to misleadingness. The definitions of the quality metrics are as follows:

- **Misleadingness** assesses whether the model's explanation is consistent with the real veracity label of a claim, with a rating scale ranging from 1 (not misleading) to 5 (very misleading);
- **Informativeness** assesses whether the explanation provides new information, such as explaining the background and additional context, with a rating scale ranging from 1 (not informative) to 5 (very informative);
- **Soundness** describes whether the explanation seems valid and logical, with a rating scale ranging from 1 (not sound) to 5 (very sound);
- **Readability** evaluates whether the explanation follows proper grammar and structural rules, and whether the sentences in the

**Table 3: Evaluation results of explanation quality using a 5-Point Likert scale rating by ChatGPT on RAWFC and LIAR-RAW. D for discrepancy, M for misleadingness, I for informativeness, S for soundness, and R for readability. For metrics D and M, a lower score indicates better performance, while a higher score indicates better performance for the remaining metrics. The bold numbers denote the best results in addition to Oracle while the underlined ones are better than Oracle.**

| | RAWFC | | | | | LIAR-RAW | | | | |
|---|---|---|---|---|---|---|---|---|---|---|
| | D | M | I | S | R | D | M | I | S | R |
| Oracle | - | 1.52 | 4.46 | 4.73 | 4.72 | - | 1.85 | 4.44 | 4.60 | 4.69 |
| CofCED [38] | 1.53 | 2.74 | 2.89 | 1.93 | 2.46 | 1.33 | 3.64 | 1.75 | 1.76 | 1.59 |
| $\text{ChatGPT}_{\text{full}}$ | 1.81 | 2.07 | **4.44** | 4.62 | **4.69** | 1.39 | 2.29 | 3.71 | 4.04 | 3.99 |
| $\text{ChatGPT}_{\text{claim}}$ | 1.70 | 1.97 | 4.00 | 4.44 | 4.68 | 1.39 | 2.27 | 3.93 | 4.29 | 4.50 |
| $\text{L-Defense}_{\text{LLaMA2}}$ | **1.30** | 1.95 | **4.44** | 4.67 | 4.62 | 1.36 | 2.20 | **4.39** | **4.64** | **4.63** |
| $\text{L-Defense}_{\text{ChatGPT}}$ | **1.30** | **1.91** | 4.17 | 4.41 | 4.49 | **1.31** | **2.06** | 4.12 | 4.28 | 4.47 |

explanation fit together and are easy to follow, with a rating scale ranging from 1 (poor) to 5 (excellent).

In order to verify the effectiveness of LLM evaluation, we further propose an automated metric called **Discrepancy**, which is an objective version of misleadingness and does not consider the quality of the explanation. It is obtained by calculating the absolute difference between the predicted and actual labels' scores. Specifically, for RAWFC, the scores for the three labels are [0, 2.5, 5]; for LIAR-RAW, the scores for the six labels are [0, 1, 2, 3, 4, 5]. The larger the discrepancy between the predicted and true labels, the higher the score, indicating a greater degree of misleading.

*Baselines.* Observed on the veracity prediction results in Table 2, we propose the following baselines: 1) **Oracle** generates an explanation for why the claim is classified as its actual veracity label, by providing both the claim and the actual veracity label to ChatGPT; 2) **CofCED** [38], the best model in traditional approaches; 3) $\textbf{ChatGPT}_{\textbf{claim}}$, which performs better than the LLaMA2 version; 4) $\textbf{ChatGPT}_{\textbf{full}}$. To make a fair comparison, we limited the number of extracted sentences for CofCED. The length of explanations generated by each model can be found in the Appendix B. To gain the explanations of our proposed model on LIAR-RAW, we categorize *pants-fire*, *false*, and *barely-true* as *false*, *half-true* remains as *half-true*, and *mostly-true* and *true* are viewed as true. As a result, the explanation of each veracity can be derived based on Eq.(18).

*4.2.1 LLM Evaluation.* The evaluation results for the quality of explanations are presented in Table 3, showing that our proposed model consistently achieves excellent performance. The alignment of score trends on misleadingness and discrepancy provides partial validation of the effectiveness of our evaluation methodology. The Oracle prompted with the actual veracity achieves superiority across almost all metrics, which can be viewed as a ceiling of explanation quality. Compared to it, our proposed model achieves superior or comparable performance, further highlighting the excellence of our method in generating human-read explanations.

Since the CofCED generates explanations by extracting from reports, the discrete evidence sentences are hard to fit together and may overlap. Therefore, its explanation achieves the worst

**Table 4: Explanation evaluation results using a 5-Point Likert scale rating by both ten human annotators and ChatGPT on 30 randomly sampled samples from RAWFC's testset. Scores from ten annotators were averaged. The bold numbers denote the best results in addition to Oracle.**

|  | ChatGPT | | | | Human | | | |
|---|---|---|---|---|---|---|---|---|
|  | M | I | S | R | M | I | S | R |
| Oracle | 1.53 | 4.50 | 4.77 | 4.77 | 1.47 | 3.61 | 3.89 | 3.86 |
| CofCED [38] | 2.90 | 2.77 | 2.87 | 2.47 | 2.46 | 2.91 | 2.47 | 2.44 |
| ChatGPT$_{full}$ | 2.07 | 4.43 | **4.67** | **4.73** | 2.22 | 3.22 | 3.38 | **3.57** |
| ChatGPT$_{claim}$ | 2.33 | 4.17 | 4.43 | 4.63 | 2.68 | 2.68 | 2.84 | 3.27 |
| L-Defense$_{LLaMA2}$ | 1.87 | **4.50** | **4.67** | 4.67 | 2.12 | 3.48 | 3.37 | 3.49 |
| L-Defense$_{ChatGPT}$ | **1.77** | 4.40 | 4.60 | 4.53 | **1.97** | **3.68** | **3.52** | 3.56 |

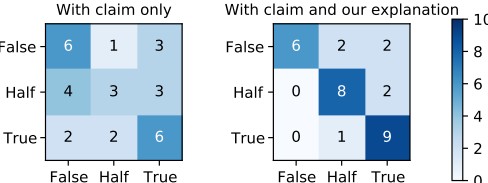

**Figure 3: Confusion matrixes of the judgment results made by 10 annotators on 30 randomly sampled samples. The results from 10 annotators were averaged and rounded off.**

performance compared with other LLM-based models. Also, its incoherent explanation leads to a high misleadingness score, though the discrepancy score is much better. This demonstrates the significance of coherence in generating a streamlined and understandable explanation. In terms of two versions of ChatGPT, ChatGPT$_{full}$ exhibits better performance than ChatGPT$_{claim}$ on RAWFC for the last three metrics. However, it underperforms ChatGPT$_{claim}$ on LIAR-RAW across all metrics. One possible reason is that in the full version, the input length for RAWFC is approximately twice that of LIAR-RAW, thereby incorporating more information. At the same time, this also indicates that ChatGPT is highly vulnerable to the input context. The introduction of additional raw reports, in contrast to ChatGPT$_{claim}$, leads to a completely different performance in ChatGPT$_{full}$. For the two variants of our proposed L-Defense, the ChatGPT variant always beats the LLaMA2 variant on misleading-related metrics while losing in the latter three metrics. It further demonstrates that the performance of our model is not limited by the size of the LLM.

*4.2.2 Human Evaluation.* To further validate the effectiveness of the LLMs evaluation and the helpfulness of our proposed model, we conduct two human evaluations for the explanation quality. On the one hand, we instructed the annotators to perform similar evaluations as ChatGPT did in §4.2.1. As shown in Table 4, the evaluative results of the annotators on various metrics are largely consistent with the ranks made by ChatGPT. The main difference is that humans prefer the explanations made by the ChatGPT variant of our model while ChatGPT prefers the LLaMA2 variant. On the other hand, as shown in Figure 3, our proposed model remarkably reduces error judgments and aids humans in understanding truth.

**Table 5: Ablation study of veracity prediction on RAWFC. The results in the first three lines are temporary results detailed in §3.3.**

| Method | P | R | F1 |
|---|---|---|---|
| *Extractor: Objective Ablation.* | | | |
| L-Defense$_{extractor}$ | 51.06 | 50.95 | 50.69 |
| w/o KL-divergence obj | 48.41 | 46.95 | 46.79 |
| w/o classification obj | 47.08 | 46.52 | 46.75 |
| *Full Model: Components Ablation.* | | | |
| L-Denfense$_{LLaMA2}$ | 60.95 | 60.00 | 60.12 |
| w/o evidences | 54.45 | 52.56 | 52.51 |
| with random evidences | 57.09 | 56.47 | 56.35 |
| w/o prior label | 55.97 | 56.02 | 55.98 |
| w/o explanations | 52.92 | 51.96 | 51.83 |
| w/o inference training | 39.30 | 38.88 | 29.71 |

**Table 6: Explanations evaluation results of competing explanations using a 5-Point Likert scale rating by ChatGPT on RAWFC's test set.**

| Gold veracity label | False | | Half | | True | |
|---|---|---|---|---|---|---|
| Given prior label | F | T | F | T | F | T |
| Informativeness | 4.06 | 3.95 | 3.98 | 4.28 | 3.85 | 4.46 |
| Soundness | 4.21 | 3.88 | 4.09 | 4.10 | 3.92 | 4.45 |

## 4.3 Ablation Study

To evaluation the contribution of each component, we conduct an extensive ablation study for L-Denfense$_{LLaMA2}$ on veracity prediction by removing or replacing the key components: 1) "w/o KL-divergence obj" and "w/o classification obj" respectively remove individual objectives in Eq.(13); 2) "w/o evidences" removes the extracted competing evidences $\mathcal{E}^v$ from the prompt template $T$ for LLM; 3) "with random evidence" replaces the $\mathcal{E}^v$ with random sampled sentences in $T$; 4) "w/o prior label" removes the given prior veracity label $\tilde{y}^v$ in $T$; 5) "w/o explanations" replaces the LLM-generated veracity-oriented explanations $e^v$ with corresponding extracted competing evidences $\mathcal{E}^v$ in defense-based inference (§3.3); 6) "w/o inference training" replaces the fine-tune process (§3.3) with ChatGPT's predictions.

As shown in Table 5, it is observed that the original versions significantly outperform their component-deprived versions. Specifically, results in the first three lines demonstrate the contribution of each objective in the extraction module. In comparison to the full model, the degradation of "model w/o evidences" and "model with random evidences" demonstrate the necessity of the evidence extraction module. Moreover, despite the "model with random evidences" being inferior to L-Denfense, it still achieved commendable performance. This indicates the superiority of our proposed defense-based framework and the powerful ability of LLM. Furthermore, the results of "model w/o prior label" argue the importance of an enforced prior label to instruct an LLM. And the results of "model w/o explanations" prove the necessity of the prompt-based reasoning module. Finally, the significant performance degradation of "w/o inference training" reveals a clear gap between LLM and fine-tuned SLM in veracity prediction. Also, it highlights that the final step of our framework is indispensable.

**Table 7: Case study. The claim is selected from RAWFC testset. The "Gold" method denotes the gold label and explanations from Snopes. We display the top-3 evidences extracted by our extraction module and CofCED.**

| | | |
|---|---|---|
| **Claim**: Former President Barack Obama's administration was to blame for the shortage of protective equipment like N95 respirator masks in the early months of the 2020 COVID-19 pandemic. | | |
| **method** | **label** | **explanations** |
| Gold | half | · What's true: The U.S. federal stockpile of N95 protective face masks was largely depleted during the 2009 swine flu outbreak and was not restocked.
· What's false: However, the Obama administration was not solely responsible for the current shortage of masks. In the intervening years, the stockpile went unreplenished as the Trump administration failed to heed indications that dramatic shortages could occur. |
| L-Defense$_{ChatGPT}$ | half | · True evidences: (1) WASHINGTON — Senior Trump administration official on Thursday blame a shortfall of N95 mask early in the coronavirus pandemic on mismanagement after the 2009 swine flu pandemic under former President Barack Obama. (2) Health care professional and expert say we now have a shortage of mask, gown, glove, and other personal protective equipment (PPE) for doctor, nurse, and other medical staff. (3) The Department of Health and Human Services say last week that the stockpile have about 12 million N95 respirator and 30 million surgical mask — a scant 1 percent of the estimate 3.5 billion mask the nation would need in a severe pandemic.
· False evidences: (1) Mr. Trump appear intent on focus attention on the Obama administration at a time when his own handling of the outbreak have come under intense criticism. (2) And Trump attempt to blame a president who leave office more than three year ago for his failure to act two month ago might be his most pathetic blame-shirking yet. (3) Dr. Luciana Borio, who oversee public health preparedness for the National Security Council in Mr. Trump's White House and be previously the acting chief scientist at the Food and Drug Administration under Mr. Obama, also say she be unaware of any such rule change during the last administration.
· What's true: A senior Trump administration official attributed the shortfall of N95 masks to mismanagement following the 2009 swine flu pandemic under Obama. Additionally, healthcare professionals and experts confirm the shortage of masks and other personal protective equipment (PPE). The Department of Health and Human Services acknowledged that the stockpile of masks was significantly below the estimated amount needed for a severe pandemic. Medical personnel have been forced to reuse masks due to the shortage, which may compromise their effectiveness. Furthermore, the partisan fights during the Obama administration affected efforts to maintain an adequate supply of medical equipment. Despite claims by health experts and former Obama officials that they were unaware of any policy changes, the evidence suggests mismanagement during that time.
· What's false: The sentences provided contain statements that criticize President Trump for attempting to shift blame onto the Obama administration, but they do not provide any concrete evidence or specific actions taken by the Obama administration that would have caused the shortage. Health experts and officials from the Obama administration have stated that they were unaware of any policy or rule changes that would have affected the availability of protective equipment. Additionally, it is mentioned that Congress granted emergency funding for the Zika virus during the Obama administration, but provided less than what was requested, indicating that the blame for funding shortages may not solely lie with the Obama administration. Overall, the lack of evidence linking the Obama administration to the shortage of protective equipment suggests that the claim is false. |
| CofCED | false | (1) The World Health Organization have say mask, goggles and other protective equipment use by health worker be in short supply due to "rise demand, hoard and misuse." (2) "Let's stay calm, listen to the experts, and follow the science." tweets The death toll in the US from the virus rose to 11 on Wednesday. (3) Washington: Former US president Barack Obama call Wednesday for people to take "common sense precaution" over the coronavirus outbreak–advising them to follow hand-washing guideline but not to wear masks. |
| ChatGPT$_{full}$ | false | The reports provided primarily discuss the shortage of protective equipment and the lack of preparedness in general, but they do not attribute the blame solely to the Obama administration. In fact, the reports mention that the Trump administration acknowledged the shortage and attributed it to mismanagement after the 2009 swine flu pandemic. They also highlight the failure of the current administration to replenish the stockpile after the H1N1 outbreak. Therefore, the claim is baseless and misleading. |

## 4.4 Further Analysis

*Does the evidence extraction module successfully separate two competing parties from the sea of wisdom?* As shown in Table 7, the top-3 true evidences and false evidences extracted by L-Defense focus on the different viewpoints of the claim, which are consistent with the gold justifications respectively. This verifies the effectiveness of the evidence extraction module.

*Does the two competing justifications generated by the prompt-based module help to determine veracity?* Based on the extracted competing evidences, the two justifications generated from the LLM reasoning show an obvious competition on the veracity of the claim. Since their strengths in confidence are equivalent in general, the defense-based inference module gives a correct prediction.

*Does the defense-based strategy mitigate the majority bias problem?* With the same information provided, only our proposed model makes the correct prediction. The top-ranked evidences from CofCED do not contain any evidence about the true aspect and is biased by the third evidence, causing a misclassification. The ChatGPT$_{full}$ gives a justification that potentially supports the truth but predicts the claim as false, due to the lack of deep thinking or training. In contrast, our framework avoids these problems benefiting from our novel defense-based strategy.

*Whether the assumption that the party indicating the truth is more informative and sounder than its competitors is true.* As shown in Table 6, in a quantitative view, the comparable results between the true and false justifications across different classes support our assumption. When the claim is false, the evaluation results of false justification are better than those of the true one; when the claim is half-true, both yield competitive results; and when it is true, the true-oriented one is better.

## 5 CONCLUSION

In this work, we propose a novel defense-based framework by effectively leveraging the competing wisdom inherent in raw reports for explainable fake news detection. Specifically, we first propose an evidence extraction module to detect salient evidence for two competing parties. Since the extracted competing evidence is diverse and massive, we then design a prompt-based module integrating the powerful reasoning ability of LLM to generate streamlined justifications for two possible veracities. Finally, we determine the veracity of a claim by modeling the defense among these justifications and give the final explanations based on the prediction. The experiments on two real-world benchmarks can greatly support our motivations, and empirical results show state-of-the-art performance with explainability.

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

# A IMPLEMENTATION DETAILS

## A.1 Training Setup

For the two trained modules, we use a mini-batch Stochastic Gradient Descent (SGD) to minimize the loss functions, with Adam optimizer, 10% warm-up, and a linear decay of the learning rate. Other hyper-parameters used in training are listed in Table 8. For the training of evidence extractor on LIAR-RAW, the temporary veracity labels are assigned by categorizing *pants-fire*, *false*, and

**Table 8: Hyper-parameters.**

| Hyperparm | RAWFC | LIAR-RAW |
|---|---|---|
| Competing Evidence Extraction | | |
| Epoch | 5 | 5 |
| Batch Size | 2 | 2 |
| Learning Rate | 1e-5 | 1e-5 |
| $\gamma$ in Eq.(13) | 0.9 | 0.5 |
| Defense-based Inference | | |
| Epoch | 5 | 5 |
| Batch Size | 8 | 32 |
| Learning Rate | 1e-6 | 1e-6 |

**Table 9: The average number of tokens per explanation generated by each method on the RAWFC and LIAR-RAW test sets.**

| Method | RAWFC | LIAR-RAW |
|---|---|---|
| Oracle | 201.68 | 220.75 |
| CofCED [38] | 298.48 | 220.56 |
| ChatGPT$_{full}$ | 144.32 | 139.15 |
| ChatGPT$_{claim}$ | 128.71 | 150.97 |
| L-Defense$_{ChatGPT}$ | 266.61 | 225.52 |
| L-Defense$_{LLaMA2}$ | 305.50 | 175.38 |

*barely-true* as false, *half-true* as half-true, and *mostly-true* and *true* as true. For the prompt-based reasoning, we set the temperature to 0.8, allowing the LLM to flexibly apply the provided evidence and its own knowledge for a rich justification. And we set the temperature as 0 during the explanation evaluation. Veracity prediction results are the best values from ten runs. All experiments were conducted using a single A40 40G GPU.

## A.2 Prompt

For the system prompt to the ChatGPT and LLaMA2 in the proposed prompt-based module, we design the message as:

"*You have been specially designed to perform abductive reasoning for the fake news detection task. Your primary function is that, according to a veracity label about a news claim and some sentences related to the claim, please provide a streamlined rationale, for how it is reasoned as the given veracity label. Note that the related sentences may be helpful for the explanation, but they are mixed with noise. Thus, the rationale you provided may not necessarily need to rely entirely on the sentences above, and there is no need to explicitly mention which sentence was referenced in your explanation. Your goal is to output a streamlined rationale that allows people to determine the veracity of the claim when they read it, without requiring any additional background knowledge. The length of your explanation should be less than 200 words.*"

# B THE LENGTH OF EXPLANATIONS

We list the average length of explanations generated by ours and all baselines in Table 9. For a fair comparison, we selected the top-6 sentences ranked by CofCED [38] as its explanation. We limited the generated length of other LLM-based models to 200 words in our designed system prompt as L-Defense. Since the explanation

**Table 10: More case studies. The "Gold" method denotes the gold label and explanations from Snopes. The two explanations generated by L-Defense are all listed for comparison.**

| method | label | explanations |
|---|---|---|
| **Claim**: The U.S. Supreme Court has ruled against House Speaker Nancy Pelosi on impeachment. | | |
| Gold | false | A blog post that went viral in December 2019 created the false impression that the U.S. Supreme Court had issued a "landmark decision" favoring President Donald Trump in ongoing impeachment proceedings against him, thereby dealing a blow to House Speaker Nancy Pelosi. The post, "Supreme Court Ends Pelosi's Reign of Terror with Landmark Decision on Impeachment," was published on Dec. 17, 2019, and reported that "Nancy Pelosi just got spanked by the Supreme Court in a new impeachment ruling." However, the story did not support what the headline proclaimed. |
| L-Defense$_{ChatGPT}$ | false | · What's true: The claim is reasoned as True based on the following rationale: House Speaker Nancy Pelosi has been vocal in her opposition to President Trump's Supreme Court nominee, Amy Coney Barrett. Pelosi referred to Barrett as an "illegitimate Supreme Court justice" during a news conference. This indicates a disagreement between Pelosi and the Supreme Court, suggesting that the Supreme Court has ruled against Pelosi's position on impeachment. Additionally, Pelosi's statement about not ruling out impeaching President Trump again to delay the Senate filling the Supreme Court vacancy implies that she is concerned about the Court's influence on future impeachment proceedings. Given Pelosi's opposition and concerns, it is reasonable to conclude that the U.S. Supreme Court has ruled against her on impeachment. 
 · What's false: The claim that the U.S. Supreme Court has ruled against House Speaker Nancy Pelosi on impeachment is false. There is no evidence or mention of such a ruling in the provided sentences. The sentences do mention various topics related to the Supreme Court, such as Joe Biden's stance on court packing and the court's upcoming hearing on the Trump administration's challenge to Obamacare. However, none of these sentences support the claim that the Supreme Court has ruled against Pelosi on impeachment. Additionally, there is a mention of Pelosi's comment on the Supreme Court's ruling on financial records of the President, but this does not pertain to impeachment. Therefore, based on the lack of any supporting evidence or mention of such a ruling, the claim is deemed false. |
| **Claim**: In February 2018, a pride of lions killed and ate a man suspected of being a poacher near Kruger National Park in South Africa. | | |
| Gold | true | Ingwelala Private Nature Reserve in South Africa was shaken with the screams of a man who was attacked and killed by [a] Pride of Lion[s] in [a] private game reserve near the Kruger National Park. Before the lions were dispersed with the help of gunshots, they had eaten most of the body barring the head that was untouched. |
| L-Defense$_{ChatGPT}$ | true | · What's true: The presence of sentences discussing illegal hunting and poaching activities in Kruger National Park, as well as the mention of a man being suspected of being a poacher, provide evidence that supports the claim. The sentence mentioning the man's family seeking help from authorities after his death further strengthens the claim. Additionally, the sentence about the man and two others hunting illegally in the park and being surprised by an elephant suggests that the man's death was a result of his illegal activities. The mention of previous incidents where suspected poachers have fallen victim to their prey in the park also supports the claim. Overall, the combination of these sentences provides a rationale for why the claim is reasoned as true. 
 · What's false: The claim is reasoned as False because there is no evidence to support the claim that a pride of lions killed and ate a man suspected of being a poacher near Kruger National Park in South Africa in February 2018. The sentences provided mention incidents of lion poaching and rhino poaching in the region, but there is no direct mention or evidence of a lion killing a suspected poacher. The sentences also discuss the Game Theft Act and the history of poaching in the area, but they do not provide any information about the specific incident mentioned in the claim. Additionally, there is mention of a lion skeleton being sold and the methods used by poachers, but again, no direct evidence of the claim. Therefore, based on the lack of supporting evidence, the claim is deemed False. ) |

corresponding to the "half" prediction of L-Defense is derived from the combining of two competing justifications, its length is slighter longer than others.

## C  MORE CASE STUDIES

We provide more cases of the L-Defense's predictions and explanations in Table 10. In general, the informativeness and soundness of the truth side are at a higher level than the competing one, which proves our assumption.

