# OpenReview forum: "Explainable Fake News Detection With Large Language Model via Defense Among Competing Wisdom"
_ACM.org/TheWebConf/2024/Conference — TheWebConf24_

### Official Review · Reviewer_EHb7 · 2023-11-24

**Novelty:** 4
**Technical Quality:** 3

**Review:**

The paper presents a novel approach for fake news detection, introducing a defense-based framework that leverages competing veracities for enhanced interpretability. The use of LLMs to generate adds a unique perspective. The proposed framework contributes to the field of fake news detection by using LLMs to provide interpretable justifications, enabling timely debunking. However, the main challenges of using LLMs to conduct explainable fake news still remain unsolved. For example, the extent of the veracity of reasoning created by LLMs is unknown, as LLMs such as ChatGPT used web pages for training. In addition, it is not convincing enough to use ChatGPT to evaluate the quality of explanations.

**Questions:**

1. AI models may confidently generate inaccurate outputs that aren't justified by their training data. Given the unknown extent of veracity created by LLMs, how might these uncertainties impact the reliability and generalizability of your proposed framework?
2. While your framework aims to mitigate majority bias, how do you measure it? Are there measures in place to handle biases effectively?
3. This paper uses ChatGPT for evaluating explanation quality. Could you provide more insights into why human judgment is not used? How do you justify how reliable ChatGPT's judgments are?
4. Given the time-sensitive nature of news, and considering that the LLMs used in your study are already pre-trained on earlier data, could you provide further details on how your approach effectively tackles the challenges posed by rapidly evolving situations and emerging events?

**Reviewer Confidence:**

4: The reviewer is certain that the evaluation is correct and very familiar with the relevant literature

**Scope:**

4: The work is relevant to the Web and to the track, and is of broad interest to the community

---

### Official Review · Reviewer_9UWi · 2023-11-24

**Novelty:** 3
**Technical Quality:** 5

**Review:**

Pros:
1. The proposed method leverages the power of the recent popular LLMs in a proper way (instead of proposing a brutal all-in-LLM solution) and has a reasonable design.
2. The analysis is comprehensive and meaningful. The analysis text is organized in a reader-friendly way.
3. The overall writing is of good quality, making this submission easy to follow (though there are still some minor typos).

Cons:
1. Regarding the remarks of existing methods:
   1) This work starts with an opinion that existing methods of explainable fake news detection lead to majority bias on "evidence". I am not very convinced by this claim. From (Yang et al., 2022), I do not find any design that shows its tendency to aggregate the major opinions. And it uses max-pooling for the selected evidence sentences, which seems irrelevant to "majority bias". It is more likely to follow the strongest signals of each dimension. This is a very fundamental issue and I expect the authors to clearly explain.
   2) The recent works using LLMs in fake news detection are considered "lacking the ability to generate an explanation." I noticed that (Hu et al., 2023) [2] that you cited acquired insights from LLM-generated rationales. Though they are side products, they could be seen as some kind of explanation. I feel the claim is not proper given such a context.
2. Regarding the comprehensiveness of the relevant work section: The relevant work section might be better with citing relevant ``old'' papers. This paper considers the conflicting opinions from different sources. This is discussed by Jin et al. in a AAAI 2016 paper [1]. I believe adding this would make the literature research more comprehensive.
3. The details about annotator quality are vacant. And we know little about how the annotators are paid (or they volunteer to do this?)
4. The failure case analysis and discussion on future works are not provided.
5. Some presentation issues exist. Please check and revise carefully. Details:
   1) L221: proposes -> propose
   2) L228: uses -> use
   3) L269: consisted of -> consisting of
   4) L12: "...which are debunking delayed and low efficiency." This seems to violate typical grammatical rules.

[1] News Verification by Exploiting Conflicting Social Viewpoints in Microblogs. AAAI 2016.

[2] Bad Actor, Good Advisor: Exploring the Role of Large Language Models in Fake News Detection. arXiv preprint.

**Questions:**

1. L271: What does the class "..." refer to? It would be better to specify at this very moment.
2. L514: What version did you use? gpt-3.5-turbo-0301? Considering that OpenAI often updates its API, the exact version number should be provided
3. In Table 2, sometimes the LLaMA2-based L-Denfense outperforms the ChatGPT-based version. Are there any comments on this?
4. For evaluation of generated explanations, were there any circumstances in which the LLM (i.e., ChatGPT) refuses to evaluate or provide invalid answers (because sometimes LLMs do not follow the numerical range rules)? If there were, how did you handle this?
5. Could you please provide the details on how you finally selected the used prompt template?
6. Is there any possibility of ChatGPT using its own knowledge to provide an explanation? How do you check the validness?

Also, it is welcome to respond to the points in the Review part.

**Reviewer Confidence:**

4: The reviewer is certain that the evaluation is correct and very familiar with the relevant literature

**Scope:**

4: The work is relevant to the Web and to the track, and is of broad interest to the community

---

### Official Review · Reviewer_K9zY · 2023-11-24

**Novelty:** 5
**Technical Quality:** 6

**Review:**

The authors propose an explainable defense-based fake news detection framework. The framework, called L-Defense works by first giving competing veracity scores to competing evidence. Next, LLM is used as a reasoner to give two prior labels and explanations. Finally, the output form LLM is used to derive final veracity label and explanation. The experiment results on two datasets show the effectiveness of L-Defense. Human evaluation is also performed to rate the explanations generated.

Strength:
•	This paper proposes an effective way to provide defense-based explainable fake news detection, which is very different from previous approaches.
•	This paper is well written and easy to understand.

Weakness:
•	The framework depends on splitting the evidence into two competing parities, which could be a limitation for claims that may not have such competing evidences.

**Questions:**

•	The reasoning part depends entirely on LLM. I am wondering whether using some more advanced prompting techniques could help.

**Reviewer Confidence:**

3: The reviewer is confident but not certain that the evaluation is correct

**Scope:**

4: The work is relevant to the Web and to the track, and is of broad interest to the community

---

### Official Review · Reviewer_gwHB · 2023-11-25

**Novelty:** 5
**Technical Quality:** 5

**Review:**

## Summary

The paper addresses the challenge of explaining and detecting fake news by proposing a novel defense-based explainable framework. Unlike existing methods that rely on neural networks and investigative journalism for veracity justification, the proposed approach introduces an evidence extraction module to analyze diverse opinions from the wisdom of crowds. A prompt-based module utilizes a large language model to generate concise justifications by considering two competing veracities inferred from salient evidences. The defense-based inference module models the defense among these justifications to determine the veracity, leading to improved fake news detection and high-quality explanations compared to state-of-the-art baselines.

## Paper Strengths

1. Introducing an evidence extraction module to divide numerous opinions into two competing parties and detect significant evidence respectively, which helps to improve the accuracy of interpretation in complex information environments.
2. Introducing a prompt-based module that uses large language models to extract salient evidence from numerous, diverse, and competing viewpoints to generate concise justifications.
3. Proposing a defense-based inference module to determine veracity via modeling the defense among the above generated justifications.

## Paper Weaknesses

1. Why does it need to be multiplied by $\partial _{j}^{v}$ in formula (8)?
2. The text contained in Table 7 is too long and can be further optimized, such as highlighting the key points.

**Questions:**

Please check weaknesses

**Reviewer Confidence:**

4: The reviewer is certain that the evaluation is correct and very familiar with the relevant literature

**Scope:**

4: The work is relevant to the Web and to the track, and is of broad interest to the community

---

### Decision · Program_Chairs · 2024-01-22

**Decision:**

Accept

**Comment:**

In this work the authors present a defense-based, explainable approach to news detection that takes into account different sides of the issue. The approach relies on competing evidence, which reviewers raise may not be present in all cases. The authors adequately addressed this concern in the rebuttal, and will need to revise this in their final version. Similarly, the discussion around majority bias could use better treatment in the text, although the discussion around this point was convincing. Overall, I echo the reviewers' sentiments that this is a novel approach to an important problem.